# Deep Learning for Robust Adaptive Inverse Control of Nonlinear Dynamic Systems: Improved Settling Time with an Autoencoder

**DOI:** 10.3390/s22165935

**Published:** 2022-08-09

**Authors:** Nuha A. S. Alwan, Zahir M. Hussain

**Affiliations:** 1College of Engineering, University of Baghdad, Baghdad 10017, Iraq; 2School of Engineering, Edith Cowan University, Joondalup, WA 6027, Australia

**Keywords:** deep learning, adaptive inverse control, robust control, nonlinear plant, autoencoder, sensor control

## Abstract

An adaptive deep neural network is used in an inverse system identification setting to approximate the inverse of a nonlinear plant with the aim of constituting the plant controller by copying to the latter the weights and architecture of the converging deep neural network. This deep learning (DL) approach to the adaptive inverse control (AIC) problem is shown to outperform the adaptive filtering techniques and algorithms normally used in adaptive control, especially when in nonlinear plants. The deeper the controller, the better the inverse function approximation, provided that the nonlinear plant has an inverse and that this inverse can be approximated. Simulation results prove the feasibility of this DL-based adaptive inverse control scheme. The DL-based AIC system is robust to nonlinear plant parameter changes in that the plant output reassumes the value of the reference signal considerably faster than with the adaptive filter counterpart of the deep neural network. The settling and rise times of the step response are shown to improve in the DL-based AIC system.

## 1. Introduction

The efficient control of nonlinear dynamic systems has been dealt with extensively in the literature. As a control problem, the essence is to make the nonlinear system output track a reference trajectory. Many approaches require accurate knowledge of the nonlinear plant in order to linearize the plant dynamics around an operating point, such as in gain scheduling [1]. Others, e.g., feedback linearization methods, require the accurate dynamic inversion of the plant [2]. These methods depend upon a reasonably accurate model or inverse model of the plant, in addition to their computational complexity. Adaptive control is the logical solution to the unknown plant control problem [3,4,5]. An adaptive filter uses its input and a desired response to modify its internal parameters such that a function of the error between its actual and desired output is minimum. Just as adaptive control of linear plants requires adaptive filtering techniques, adaptive control of nonlinear plants requires nonlinear adaptive filtering methods, which are generally achieved by using neural networks (NN) [6]. 

In the present work, adaptive inverse control (AIC) will be considered, as it is particularly simple to implement. AIC was first devised by Widrow and Walach [7] as a result of approaching the discipline of adaptive control from the point of view of adaptive signal processing. A basic schematic of linear systems is shown in Figure 1, in which an adaptive transversal filter approximates the inverse of the unknown plant when connected with the latter in an inverse modeling setting [3]. This adaptive filter is copied to form the controller in the figure. During normal operation, the plant output will track the reference signal, which is input to the controller, the output of which is the driving signal to the plant. Delay $z^{-\Delta }$ makes up for possible internal delays. The control aim is to guarantee that the reference signal can be tracked accurately with minimum delay. Ideally, the cascade of the controller and plant forms a unit-magnitude transfer function enabling the plant output to be an exact but delayed version of the reference in case of no noise and/or disturbance. The plant must be stable and with minimum phase for its inverse to be stable.

Disturbance rejection continues to constitute an important issue in process control [8]. Generally, it eliminates unwanted inputs that would otherwise affect the output and result in system error. For AIC systems, methods exist for controlling disturbances and plant noise without compromising the control of plant dynamics [6,7]. As these two control procedures are easy to separate, these methods will not be considered further in the present work, whose focus is on nonlinear adaptive control strategies. 

It is important to note that this is feed-forward control, since no feedback is employed except that which is inherent in the adaptive algorithm. This adaptive algorithm could be, for example, the least mean square (LMS) algorithm [3] for the linear adaptive control case. When the plant is nonlinear, a nonlinear adaptive filter that can be realized by a neural network is needed. The adaptive algorithm would then be the back-propagation (BP) algorithm, for example [9]. The updating of the adaptive inverse model in Figure 1 enables real-time control (RTC) to continuously track any parameter change in the plant, resulting in robustness to parameter change. Otherwise, a non-adaptive neural inverse model control system would be very sensitive to parameter changes [10]. As hinted at in [11], RTC can also handle time-varying or predicted parameter changes via triggered (possibly periodically) actions to activate updating. This procedure could markedly reduce the computational burden required for continuous updating or tracking. 

In [6,12], AIC was successfully performed for linear and nonlinear systems using shallow neural networks. It is known that any smooth function (linear or nonlinear) can be approximated by a shallow neural network with a sufficient number of neurons in the hidden layer. Using deeper networks with more than one hidden layer, however, results in more efficient function approximation, since deep neural networks with standard architectures represent the compositionality of functions [13]. Deep neural networks with more than one hidden layer could not be trained successfully without the deep learning (DL) techniques arrived at in the mid-2000s. These techniques include the use of rectified linear units (ReLU) as activation functions, the use of the dropout technique during training and the advent of efficient computing hardware such as graphical processing units (GPU) [14].

In [10], the AIC of nonlinear systems was implemented using a NN with two hidden layers and sigmoidal activation functions. In contrast, the present work uses up to four hidden layers with ReLU activation functions to avoid the vanishing gradient problem that would occur if sigmoidal activation functions were used in deep NNs. 

A similar problem is treated in [15] but with non-adaptive inverse control. Recently, Lyapunov-based DL adaptive online control of nonlinear systems, as opposed to AIC, was carried out [16]. Specifically, a DL controller uses a restricted Boltzmann machine (RBM) to initialize the weight values and Lyapunov stability method to update a two-hidden-layer NN controller connected to a nonlinear plant in a negative feedback control loop. AIC, on the other hand, involves open loop control and is different from the negative feedback control described in [16]. 

Recently, deep NN-based adaptive control methods of nonlinear systems to track time-varying trajectories in real time have been proposed and are worth mentioning, although they do not deal with AIC. In [17], a real-time deep NN adaptive control architecture of nonlinear systems is proposed to simultaneously update the weights of multiple layers of the DL controller in real time; robustness against parametric uncertainties was also reported. Similar works [18,19,20] developed DL controllers containing both real-time online and offline iterative learning components. 

The aim of the present work is to achieve simple yet robust adaptive control of a generally unknown nonlinear plant that is assumed to be stable and invertible using deep NNs as controllers with dynamics that are the inverse of those of the plant. Since the theory is still being developed to guide the analysis of control systems with nonlinear plants, one of the main objectives of this work is to simply show how nonlinear AIC using DL differs from its linear counterpart when both are used to control a nonlinear plant. It will also be demonstrated that the deeper the NN controller, the more robust the control system is to parameter change. The DL controller was also shown to be superior to a shallow NN controller. To the best of the authors’ knowledge, DL with multiple deep layers has not been used with AIC in the existing literature. The assumption of the stability and invertibility of the nonlinear plant may be considered as a limitation of the present approach. 

The organization of the paper is as follows. Section 2 introduces general mathematical models for nonlinear discrete-time plants. Section 3 explains deep neural networks and their training using the BP algorithm. The DL-based AIC nonlinear control system is illustrated and explained in Section 4, and simulation results are presented and discussed in Section 5 and Section 6. Finally, Section 7 concludes the paper. 

## 2. Dynamic Nonlinear Discrete-Time Plants

Four models of discrete-time plants governed by nonlinear difference equations are described in [21] and are summarized below. The input and output of the plant are denoted by u(k) and c(k) respectively, where *k* is the discrete time index. It is assumed that the present output sample is dependent on *n* past outputs and *m* present and past inputs, where m≤n. Such a system is said to be dynamic, in contrast to a static or memory-less system. In the following model equations, f(·) and g(·) are nonlinear functions and *α**_i_* and *β**_i_* are constants. 

Model I:$$c\left(k\right)=\sum_{i=1}^{n}\alpha_{i}c\left(k-i\right)+g\left[u\left(k\right),\;u\left(k-1\right),\;\ldots ,\;u\left(k-m+1\right)\right]$$

Model II:$$c\left(k\right)=f\left[c\left(k-1\right),\;c\left(k-2\right),\;\ldots ,\;c\left(k-n\right)\right]+\sum_{i=0}^{m-1}\beta_{i}u\left(k-i\right)$$

Model III:c(k)=f[c(k−1), c(k−2), …, c(k−n)]+g[u(k), u(k−1), …, u(k−m+1)]

Model IV:(1)c(k)=f[c(k−1), c(k−2), …, c(k−n); u(k), u(k−1), …, u(k−m+1)]

The nonlinear plant functions f(·) and g(·) are assumed to be unknown when adaptive inverse control of the plant is performed. It was found in [21] that Model II is particularly suitable for control problems. If the nonlinear plant is bounded-input-bounded-output (BIBO) stable, its model must also possess this property. For Model I, this implies that the characteristic equation, comprising a polynomial in *z* of degree *n* with *α_i_* as the coefficients, must have all its roots inside the unit circle in the *z*-plane. For the other models, the stability conditions are not so simple, and therefore, they constitute an important research area. 

## 3. Deep Neural Networks

Multi-layer neural networks can be regarded as nonlinear maps with their weights as parameters [21]. They can thus be used as subsystems to constitute controllers for nonlinear dynamical plants. In the present case, the NN or the controller models the inverse of the plant dynamics, as explained in the introduction and Figure 1. Dynamics are introduced in the NN via tapped delay lines (TDL) at the input. A multi-layer NN with more than one hidden layer is considered deep [14]. The weights are adapted to minimize a function of the error between the NN output and the desired output according to a training algorithm such as the BP algorithm, which is based on the steepest descent. The training and convergence of NNs are heavily dependent on the initial conditions of the weights, as well as the nature of their input data. In this work, neural networks will be denoted by the notation ${Ɲ}_{\left(I,\;L\right):J:K:\ldots }$, where *I* is the number of input nodes, *L* is the number of output nodes (or neurons), J is the number of neurons in the first hidden layer, K is the number of neurons in the second hidden layer, and so on. A deep NN with two hidden layers is illustrated in Figure 2. The NN in this figure generally represents a multiple-input multiple-output (MIMO) system. The outputs of all hidden and output nodes are subjected to nonlinear activation functions. Only the output nodes are permitted to have a linear or nonlinear activation function depending on the application and the required range of the NN outputs. The output vector Y can be expressed as:(2)$$Y_{L\times 1}=\psi \left[G_{L\times K}\;\Phi \left\{H_{K\times J}\;\Phi \left(W_{J\times I}\;X_{I\times 1}+B_{J\times 1}\right)+\beta_{K\times 1}\right\}+b_{L\times 1}\right]$$
where X is the input vector; W, H, and G are the weight matrices of the first hidden, second hidden and output layers, respectively; I, J, K and L are the numbers of nodes of the input, first hidden, second hidden and output layers, respectively; and B, β and b are the corresponding biases. The activation functions in Equation (2) operate pointwise on the respective vectors. The BP algorithm, which is based on gradient descent, is summarized by the following set of equations, with reference to Figure 2. It is assumed that nonlinear activation functions are present in the hidden layers, whereas the output nodes are linear. 

The BP adaptation equations are given by the following set of equations:$$g_{lk}\;\leftarrow \;g_{lk}+\Delta g_{lk},\;l=1,\cdots ,L;\;k=1,\cdots ,K.$$
where
$$\Delta g_{lk}=\alpha \left(d_{l}-y_{l}\right)y_{k}=\alpha \delta_{l}y_{k}$$
$$h_{kj}\;\leftarrow \;h_{kj}+\Delta h_{kj},\;k=1,\cdots ,K;\;j=1,\cdots ,J.$$
where
(3)$$\Delta h_{kj}=\alpha \delta_{k}y_{j},\;\mathrm{with}\;\delta_{k}=\left[\sum_{l}g_{lk}\delta_{l}\right]\cdot \;\Phi^{\prime }\left(v_{k}\right)$$
$$w_{ji}\;\leftarrow \;w_{ji}+\Delta w_{ji},\;j=1,\cdots ,J;\;i=1,\cdots ,I.$$
where
$$\Delta w_{ji}=\alpha \delta_{j}x_{i},\;\mathrm{with}\;\delta_{j}=\left[\sum_{k}h_{kj}\delta_{k}\right]\cdot \;\Phi^{\prime }\left(v_{j}\right)$$

In Equation (3), α is the BP learning rate, the ds are the correct outputs at the output layer needed for supervised training, the vs are the activation function inputs, the ys are the activation function outputs for output and hidden layers, the xs are the inputs to the input layer, and $\Phi^{\prime }\left(\cdot \right)$ is the derivative of the nonlinear activation function Φ(·). In the present application, the training data, consisting of the correct outputs (d) and inputs (x), would be values of the control signal u(k) and the plant output signal c(k), respectively. 

Training deep neural networks is usually a difficult task due to the different gradient magnitudes in lower and higher layers, the curvature of the objective or error function with its numerous local minima, and the lack of acceptable generalization caused by the large number of parameters to be updated. The following sub-section highlights the use of NN autoencoders (AE) to pre-train deep NNs. 

### 3.1. Deep Neural Network Initialization Using Autoencoders

Well-designed random weight initialization of a deep NN is crucial for proper convergence during training and subsequent operation. Poorly initialized networks are very difficult, if not impossible, to train. To initialize the deep NN weights, each layer can be pre-trained separately using an auxiliary error function before the whole network is trained by the usual methods, such as stochastic gradient descent, that employs back-propagation. This method of unsupervised pretraining of one layer at a time avoids the difficulties of full deep NN supervised training. The NN architecture adopted to train each layer separately is called an autoencoder (AE). NN autoencoders have their own inputs as correct outputs during training [22,23,24]. In this work, we will use simple NN autoencoders that have only one hidden layer. To limit the ensuing computational complexity, the present work will not consider deep AEs. The procedure is as follows [24]: the NN initial weights are determined by first greedily training a sequence of shallow AEs one layer at a time using unsupervised data. A shallow AE consists of an encoder and a decoder. Only the encoder weights of the trained AE are retained and considered to be the initial weights of that layer. The final layer is trained using supervised data. Finally, the complete network is fine-tuned using supervised data according to the back-propagation algorithm. AEs not only improve system initialization; they also lead to better generalization and prevent overfitting [25,26]. 

Classical momentum is another necessary technique to improve deep NN training. Even well-initialized networks can perform poorly without momentum. This concept is explained below. 

### 3.2. Training with Momentum

Stochastic gradient descent can be accelerated by iteratively accumulating a velocity vector in the direction of reduction of the error function. If we call the velocity vector v, a weight update of Equation (3) will change from
$$w_{ji}\;\leftarrow \;w_{ji}+\Delta w_{ji}$$
to:$$v\;\leftarrow \;\mu \;v+\Delta \;w_{ji}$$
(4)$$w_{ji}\leftarrow \;w_{ji}+v$$
where μ is the momentum factor, assuming a zero initial value of v. The acceleration of the gradient descent algorithm using momentum does not come at the expense of stability, contrary to that by increasing the learning rate α. Setting μ to zero in Equation (4) reduces the equation to its original form. For convex objective functions, the momentum-based method will outperform the SGD-based method, particularly in the transient stage of optimization, and is capable of accelerating directions of low-curvature in the objective or error function [22].

## 4. DL-Based Adaptive Inverse Control of a Nonlinear Plant

Several assumptions are made to achieve satisfactory control of nonlinear plants. For example, it is assumed that the nonlinear plant is BIBO stable. They are also assumed to be non-minimum-phase, meaning that they have stable inverses, and first of all, that such inverses exist. A nonlinear system cannot always be assigned poles and zeros as conceived in linear system theory, in which a minimum-phase system has all zeros inside the unit circle and, therefore, has a stable inverse. (Note that a nonlinear system could have a linear component, as in Models I and II of Section 2). However, the term minimum-phase was used in [6] for nonlinear systems, meaning that they have stable inverses. Figure 3 shows a block diagram of the AIC system for controlling nonlinear plants using deep NNs. In contrast to negative feedback closed-loop control, and although the deep NN output affects the weight adaptation rules (Equation (3)), and hence the controller, this is feedforward control, as mentioned in the introduction. 

As explained in Section 3, weight initialization of the deep NN is achieved with the aid of pre-trained AEs. To train AEs before normal system operation, we need the input-correct output training data of the deep NN. However, in the present application, the correct output is not readily available due to the adaptive nature of this control system. Rather, the correct output builds up gradually during online operation. A solution to this problem is to use offline measurements as training data for the AEs by inputting training data to the unknown nonlinear system and measuring its output. Since the deep NN is meant to inverse-model the nonlinear system or plant, the measured plant output is used as input training data to the NN, and the chosen input data will constitute the NN correct output. All this is done offline and the NN is thereby pre-trained using AEs. 

It will be shown that the reference command signal is well tracked for this adaptive system, rendering it robust to parameter variations, especially when the NN is deep enough. If a parameter of the nonlinear plant changes, causing a fall (rise) in the plant output, the adaptive process compensates for this, causing the control signal to increase (decrease) to re-attain the original plant output that tracks the reference. The plant parameter change could be abrupt, due, for example, to a failure of the actuators or sensors [27,28]. Soft faults such as actuator or sensor biases are also frequently encountered in industry [28]. If the above system is a temperature control system, as in [10] for instance, and if the internal fan actuator abruptly changes speed to a higher value due to soft faults or failures, the temperature will drop and so will the plant output voltage. The control signal will then increase such that the system adaptively returns to its original state, and the faulty speed change of the fan has no effect on the tracking performance. Such abrupt parameter variations will be simulated and discussed in the following section, as situations of this nature are more challenging than slowly varying parameter changes, as far as tracking in adaptive control systems is concerned. 

## 5. Simulation Results and Discussion

All simulations are carried out in MATLAB (Version R2021a, Update 3, The Mathworks, Natick, MA, USA, academic licenses 30904939 and 40635944). The suggested nonlinear AIC system of Figure 3 is simulated and the results are presented. The deep NN controller is then compared to a linear controller to show the difference in performance. The nonlinear plant to be controlled is governed by the nonlinear difference equation given below, in accordance with Model II of Equation (1).
(5)c(k)=f[c(k−1), c(k−2)]+u(k)
where the nonlinear function f(·) is substituted for as follows:(6)$$c\left(k\right)=\frac{c\left(k-1\right)\;\cdot \;c\left(k-2\right)}{1+{\left[c\left(k-1\right)\right]}^{3}}+u\left(k\right)$$

Nonlinear plants of this form were introduced in the literature by Narendra and Parthasarathy [21]. The reference signal is taken as a square wave spanning 1000 samples. The NN used is ${Ɲ}_{\left(4,\;1\right):5:5:5:5}$, indicating four hidden layers, with each having five nodes with ReLU activation functions, one linear output node and an input TDL length of 4. The BP learning rate is α = 0.02. BP minimizes the quadratic cost function. As in Figure 3, the controller is a copy of the approximate inverse plant model achieved by the adaptive neural network. Figure 4a shows the square reference and plant output signals. Figure 5 shows the corresponding control signal. It is clear that with each step change, the NN re-adapts, allowing the plant output to track the reference command signal, which is applied as input to the controller. The response to the step changes is oscillatory with overshoot as shown. Apart from the adaptation region near the step changes, the plant output and reference signals are indistinguishable from each other. This is also verified from Figure 4b which shows the error curve between the reference and the plant output. Figure 6 and Figure 7 show the reference, plant output, error signals, and the control signal respectively, when an abrupt plant parameter change occurs at the 600th sample. The parameter change is assumed to render the following plant equation:(7)$$c\left(k\right)=\frac{c\left(k-1\right)\;\cdot \;c\left(k-2\right)}{1+4\;\cdot {\left[c\left(k-1\right)\right]}^{3}}+u\left(k\right)$$

A unity parameter in the denominator is then increased to 4. This sudden change will cause an abrupt but temporary decrease in the plant output and an increase in the control signal at the 600th sample, as is clear from Figure 6 and Figure 7. The settling time needed after parameter change is 18 time-samples only. The settling time is taken as the time needed for the plant output to reach and stay within 2% of its final value. The parameter changes have little effect on the tracking performance; they are rapidly compensated for. 

A further parameter change could result in the following plant equation:(8)$$c\left(k\right)=\frac{c\left(k-1\right)\;\cdot \;c\left(k-2\right)}{1+4\;\cdot {\left[c\left(k-1\right)\right]}^{3}}+2\;\cdot \;u\left(k\right)$$

The above parameter change, when effected abruptly, will cause an abrupt but temporary increase in plant output and, therefore, a decrease in the control voltage at the 600th sample as in Figure 8 and Figure 9. Again, the change is compensated for, and good tracking is obtained. 

To further demonstrate the benefit of using DL, we operate the same nonlinear control system (Figure 3) but using an adaptive FIR filter in place of the deep NN. The adaptive FIR filter uses a four-tap TDL, which is the same TDL length used in the deep NN. The results are illustrated by Figure 10 and Figure 11. The parameter change at the 600th sample is in accordance with Equation (7). The settling time is 58 time-samples when using an LMS learning rate of 0.02 as in the BP training of the deep NN. This settling time is considerably higher than that obtained with DL, a result that is to be anticipated because of the linear structure of the inverse model after convergence. The linearity implies the presence of structural errors in modeling (or inverse modeling) the nonlinear system [29]. The advantage of using DL comes at the expense of some additional computational complexity. The number of multiplications needed with DL is $O\left\{N^{2}\right\}$ with N denoting the average number of nodes per layer, whereas that needed with the adaptive filter is O{N} only. 

NN training methods based on gradient descent may cause the solution to converge to a local rather than the global minimum. As a first step, and for each comparison between systems or between plant parameter variations, initial conditions for the weight values were adjusted to yield the best performance expected from the nonlinear system when a specific scenario is to be compared with others. This is done by trial and error to discover the general trend or behavior which shortly becomes manifest. 

It is worth noting that the compensating change in control voltage upon parameter change, demonstrated by Figure 7, Figure 9 and Figure 11, can be used in control applications such as temperature measurements using thermoresistive sensors [30]. The integration of sensors in control and automation draws on several sensor functions that are featured in sensor networks, fault tolerant control, intelligent sensors, robot sensing, etc. [31].

However, in the temperature measurement application described in [30], the thermoresistive sensor itself is the nonlinear plant whose process variable (temperature) is to be controlled and fixed to a reference value, enabling the measurement of the surrounding temperature. The resistance of such a sensor depends on temperature, thereby changing the electrical signal associated with it, which is fed back and compared with the reference. To measure the surrounding temperature, the role of feedback control is to keep the sensor temperature constant and equal to a reference temperature despite the temperature change in the surroundings. Therefore, the resulting variation or change in the control voltage is used to measure the surrounding temperature. The DL-based AIC control system of Figure 3 can be a suitable substitute for the feedback control used in this sensor control problem of temperature measurement; the justifications are ease of implementation and avoidance of techniques used with nonlinear plant control such as feedback linearization. 

Table 1 shows the dependence of the settling time after parameter change, in time samples, on the number of hidden layers without and with AE initialization. The case of a shallow NN (single hidden layer) has also been included. In AE training, it is common practice to employ a BP that minimizes the quadratic cost function, using ReLU activations in the nodes [25]. As explained in Section 4, offline measurements to train the AEs are obtained by measuring input and output of the nonlinear plant. The input to the plant will constitute the correct output, and therefore it can be chosen to resemble the control signal obtained in Figure 5. It can simply be chosen as a square wave (free of transients) with step values comparable to those of the figure. Then, the measured plant output will constitute the input training data of the AEs.

In reference to Table 1, without AE, two and three hidden layers show similar performances that are intermediate between the case of four hidden layers and that of the shallow NN. The longest settling time is the result of using the adaptive FIR filter as a controller. When AEs are used for initializing the deep NN, better results are obtained regarding settling time, as can be seen from Table 1. This improvement is due to proper weight initialization, which speeds up training by guaranteeing first that the stochastic gradient solution is close to a suitable local minimum. The symbol *α*_ae_ is used to denote the AE learning rate.

Table 2 demonstrates the benefit of using a momentum factor μ on the rise time of the step response after parameter change. The rise time of a step response is defined as the time needed for the response to rise from 10% to 90% of the final value. It is found that training with a momentum factor reduces the rise time but increases the settling time without causing instability.

For control systems with underdamped step response (which is often encountered in this work), the rise time is sometimes defined as the time required by the response to reach the final value during its first cycle of oscillation. The smaller rise time due to the use of momentum is clear from Table 2. This is important in control applications; if the rise time is too long, the system may be operating with the process variable below the optimum for too long. This may have consequences, depending on the control application. For example, the consequence could be the failure to apply sufficient braking force quickly enough. 

Finally, it can be seen from Table 2 that the settling time, when using a shallow NN with momentum, is greater than that without momentum as argued, but it is smaller (better) than the cases of two and three hidden layers using momentum. (The case of four hidden layers is still superior.) Such casual deviation from the general trend can be encountered as NNs, and generally DL, are sometimes prone to inexplicable behavior.

## 6. Final Remarks

In general, adaptive control systems are robust, simply because they are adaptive in the sense that the controller weights adapt to compensate for the change in the system. When involving deep NNs, it is difficult, if not impossible, to carry out a formal analysis to prove robustness of this DL-based control system to parameter changes. The reason is that “the weights are nested within a collection of nonlinear activation functions [17]”. Deep NNs are better at function approximation (finding plant model and inverse model) than shallow ones because the former ones can learn compositionality of functions. A mathematical analysis to prove that the more intricate the nonlinear function compositionality, the more robust the control system, would not be mathematically tractable. This result can only be arrived at conceptually by reasoning that better function approximation would lead to better and faster adaptation and hence to better compensation for parameter changes, and ultimately robustness. Although backed by mathematics and statistics, DL is also, in part, an empirical branch of artificial intelligence, and many of the best performing DL algorithms have no completely theoretical proofs.

It is also noteworthy that device saturations do not affect performance as markedly as in linear control systems. Saturation arises for example from an operational amplifier whose output voltage limit is determined by the supply voltage. This and other forms cause controller saturation or saturation in any other part of the control system. This problem disables control in linear control systems [32]. However, in nonlinear control, which is the case at hand, the controller saturation phenomenon is itself a nonlinearity and should not significantly affect the controller performance. This is especially true when the nonlinear activation functions in the NN controller nodes are the sigmoidal or tanh functions that are of a saturation nature themselves. A saturation effect in an ReLU node would just change it to another form of nonlinearity. In many DL problems, the choice of the type of nonlinear activation function is more or less flexible. 

## 7. Conclusions

In this work, DL-based adaptive inverse control of nonlinear dynamic systems is achieved. Simulations indicated efficient online (tracking) control. It is evidently possible to generate a control signal that ensures reliable inverse modeling and control of a nonlinear plant with unknown dynamics using a deep neural network to learn the inverse model. The resulting adaptive control system is robust to parameter changes. The settling and rise times of the step response after parameter change are shown to improve even further when using autoencoder initialization of the neural network weights and including momentum in the cost function minimization by backpropagation; the deeper the NN that learns the inverse model, the more accentuated the robustness of the adaptive control system to parameter change. The effectiveness and favorable results of the proposed DL controller approach in adaptive inverse control systems with nonlinear plants has been demonstrated by simulation comparisons with a shallow NN controller and with an FIR adaptive filter controller. 

## Figures and Tables

**Figure 1 sensors-22-05935-f001:**
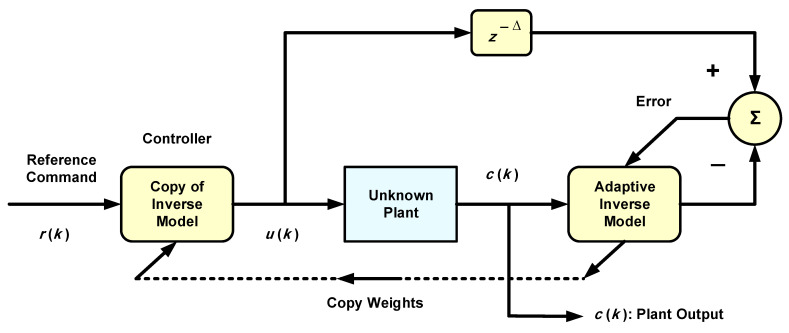
Adaptive inverse model control of an unknown plant [3].

**Figure 2 sensors-22-05935-f002:**
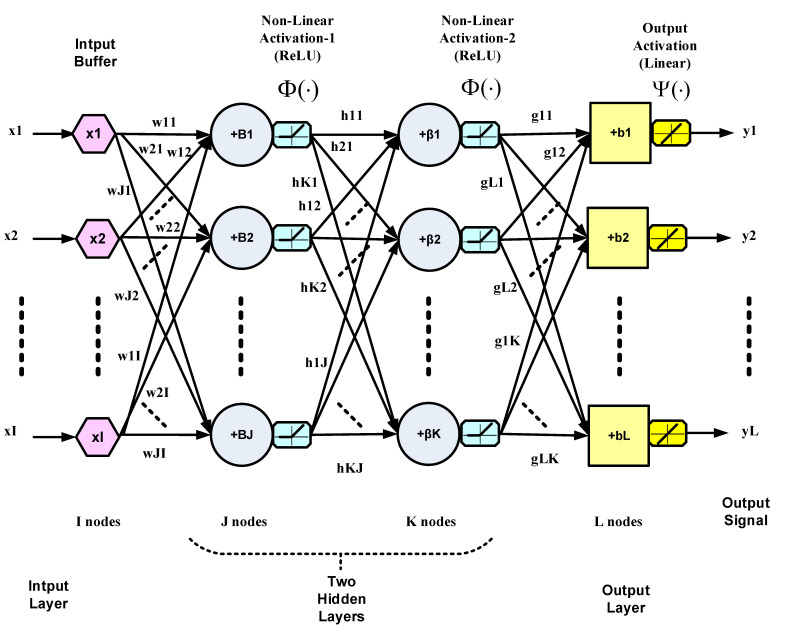
A generic diagram for a three-layer deep NN with two hidden layers.

**Figure 3 sensors-22-05935-f003:**
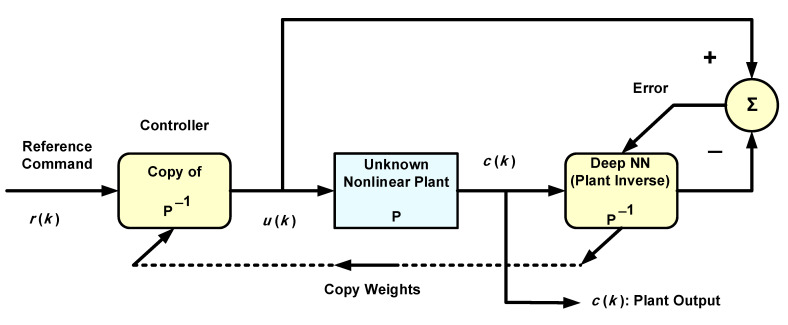
DL-based adaptive inverse control of a nonlinear plant.

**Figure 4 sensors-22-05935-f004:**
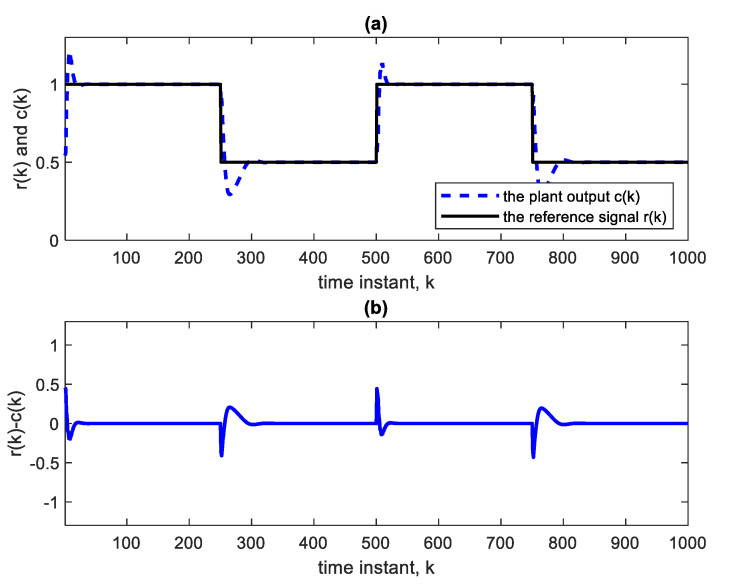
(**a**) Reference and plant output of the AIC system with ${Ɲ}_{\left(4,\;1\right):5:5:5:5}$. (**b**) Error curve r(k)-c(k).

**Figure 5 sensors-22-05935-f005:**
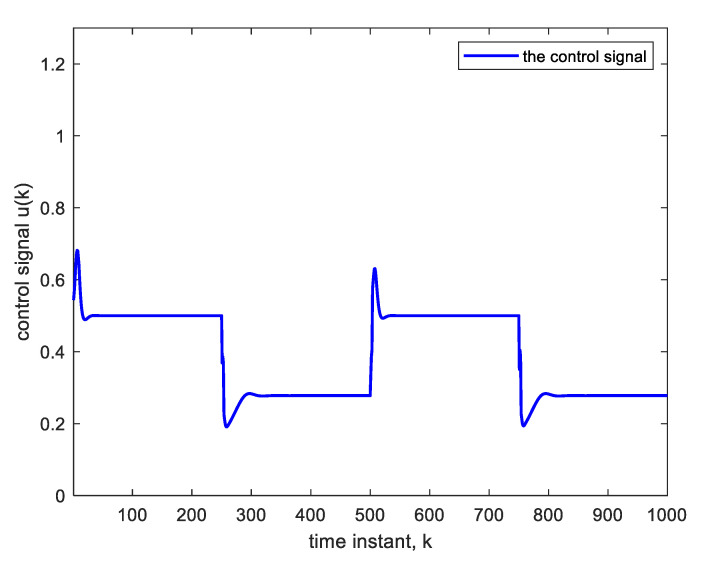
Control signal corresponding to Figure 4.

**Figure 6 sensors-22-05935-f006:**
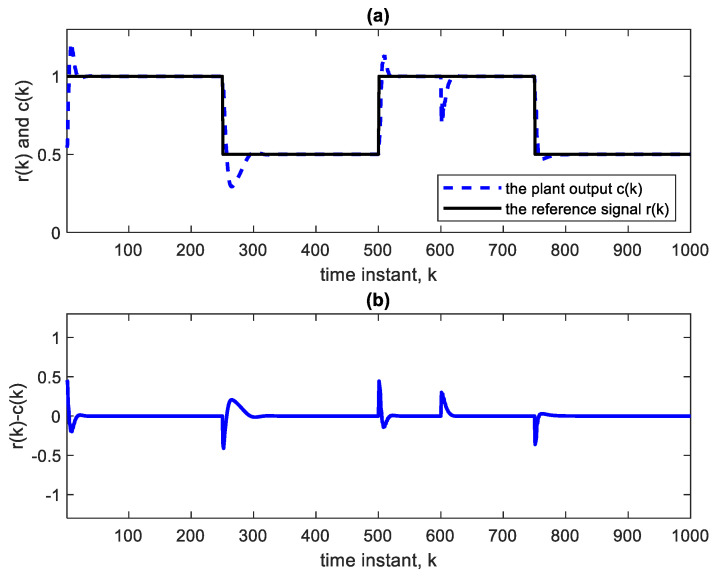
(**a**) Reference and plant output with a parameter change at the 600th sample according to Equation (7), using ${Ɲ}_{\left(4,\;1\right):5:5:5:5}$. (**b**) Error curve r(k)-c(k).

**Figure 7 sensors-22-05935-f007:**
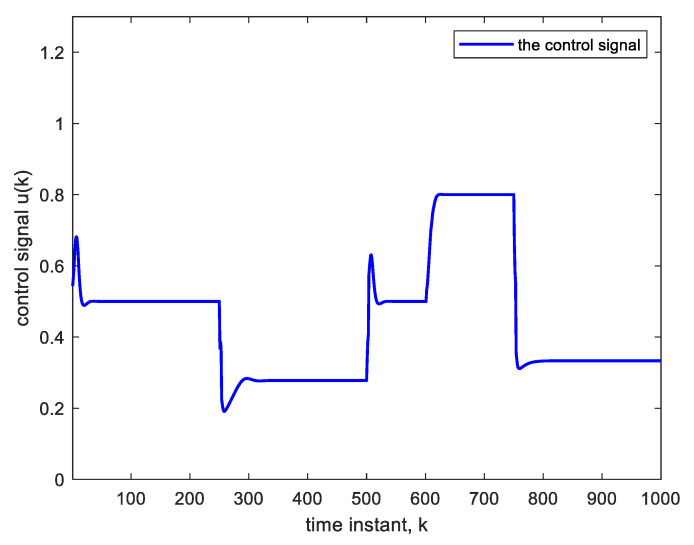
Control signal corresponding to Figure 6.

**Figure 8 sensors-22-05935-f008:**
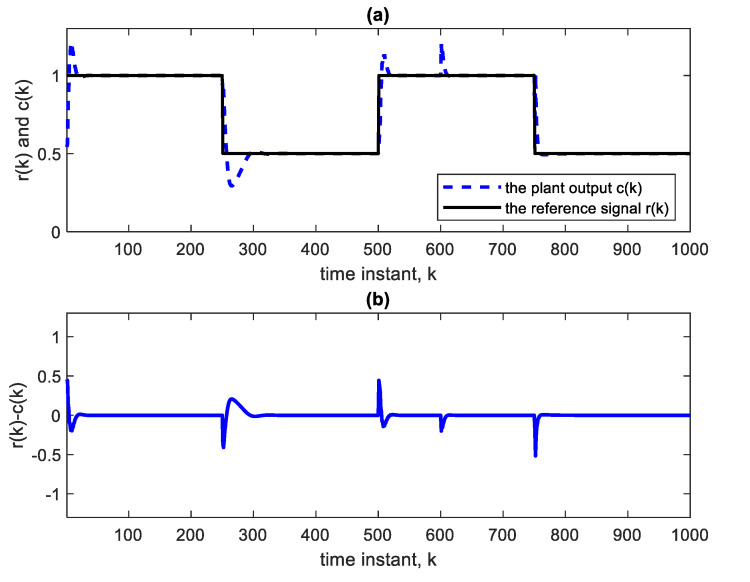
(**a**) Reference and plant output with a parameter change at the 600th sample according to Equation (8), using ${Ɲ}_{\left(4,\;1\right):5:5:5:5}$. (**b**) Error curve r(k)-c(k).

**Figure 9 sensors-22-05935-f009:**
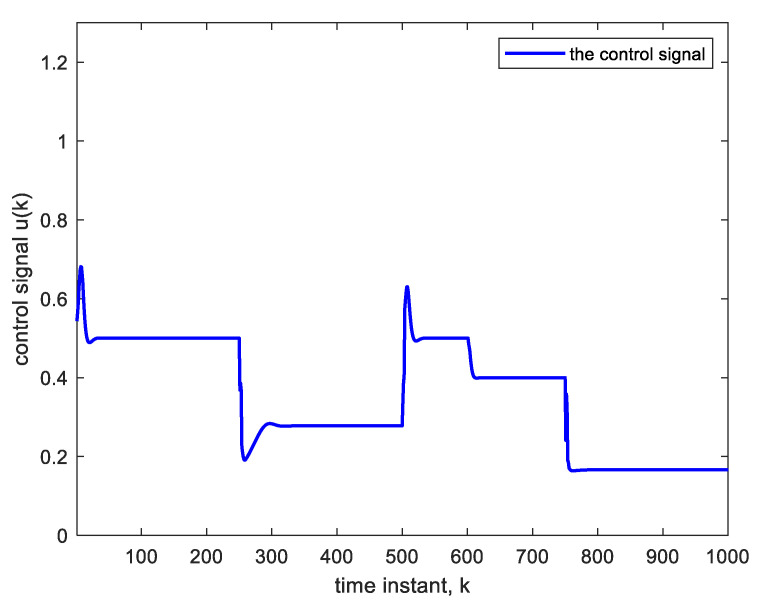
Control signal corresponding to Figure 8.

**Figure 10 sensors-22-05935-f010:**
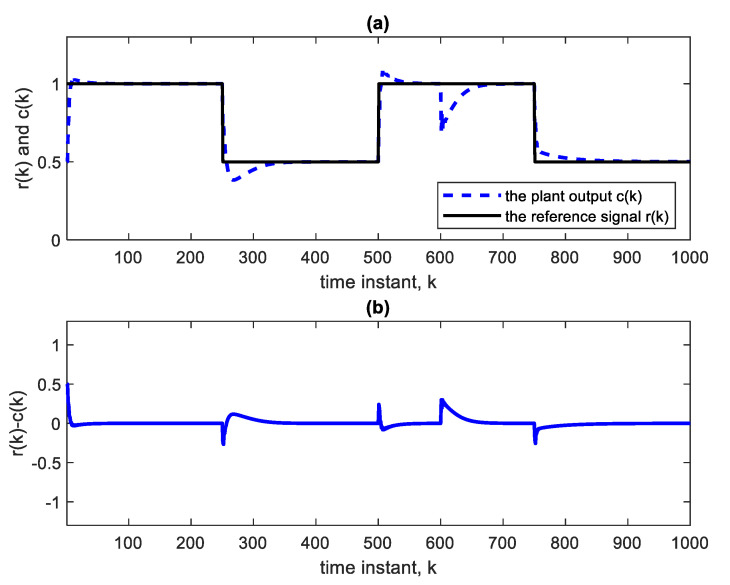
(**a**) Reference and plant output with a parameter change at the 600th sample, using adaptive filter as the inverse model of the nonlinear plant. (**b**) Error curve r(k)-c(k).

**Figure 11 sensors-22-05935-f011:**
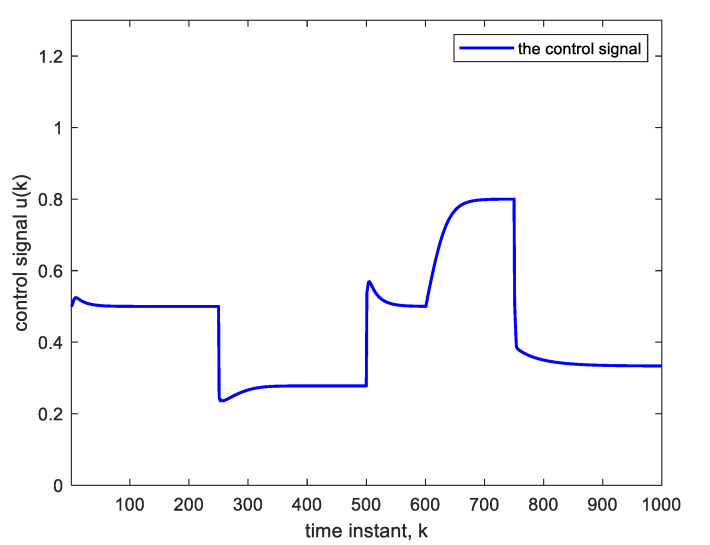
Control signal corresponding to Figure 10.

**Table 1 sensors-22-05935-t001:** Settling time after nonlinear-plant parameter change (Equation (7)) vs. number of deep NN hidden layers, and settling time when using a linear FIR adaptive filter. α = 0.02, *α*_ae_ = 0.1, μ = 0.

Type of Controller		Settling Time (In Number of Sample Intervals)	
		without AE Initialization	with AE Initialization
Deep NN	${Ɲ}_{\left(4,\;1\right):5:5:5:5}$ (4 hidden layers)	18	10
	${Ɲ}_{\left(4,\;1\right):5:5:5}$ (3 hidden layers)	25	13
	${Ɲ}_{\left(4,\;1\right):5:5}$ (2 hidden layers)	25	17
Shallow NN		30	19
Linear FIR adaptive filter		58	

**Table 2 sensors-22-05935-t002:** Momentum effect on settling time and rise time after nonlinear-plant parameter change (Equation (7)) vs. number of deep NN hidden layers, and when using a linear FIR adaptive filter. *α* = 0.02, *α*_ae_ = 0.1, different momentum factor values (μ). AE is used for initialization.

Type of Controller		Settling Time (In Number of Sample Intervals)		Rise time (In Number of Sample Intervals)	
		μ = 0	μ = 0.4	μ = 0	μ = 0.4
Deep NN	${Ɲ}_{\left(4,\;1\right):5:5:5:5}$ (4 hidden layers)	10	18	9	7
	${Ɲ}_{\left(4,\;1\right):5:5:5}$ (3 hidden layers)	13	30	12	9
	${Ɲ}_{\left(4,\;1\right):5:5}$ (2 hidden layers)	17	30	15	11
Shallow NN		19	20	17	12
Linear FIR adaptive filter		58		50	

## Data Availability

All types of data were generated using mathematical equations.

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
