# Peer review of "Deep Learning for Robust Adaptive Inverse Control of Nonlinear Dynamic Systems: Improved Settling Time with an Autoencoder"

_sensors, 2022, doi:10.3390/s22165935_

Round 1

Reviewer 1 Report

The robust adaptive control using deep neural networks for nonlinear systems is proposed in this study. The article is well-organized and  clearly written. However, improvements shall be done before publishing. Specific comments are following below.

1. At line 96, authors claim that the proposed control method is feed-forward or open loop control, since no feedback is employed to the controller. However, this is somewhat confusing because the output is used in adaptive algorithm and the parameters of controller is affected by this algorithm. In my opinion, it is a feedback control because the output affects the controller. An exact expression is required.

2. Is this algorithm for general unknown nonlinear systems or stable unknown nonlinear system? At line 98-99, it is introduced that the proposed algorithm is proposed for generally unknown nonlinear systems. But, at line 131-135 and line 230-231, the stable systems are considered. Which one is correct? The target system should be clarified.

3. Are there any previous or similar approaches that are using deep learning and adaptive filtering techniques? Relevant research needs to be supplemented.

4. At line 87, 90, 95, 99, and 104, 'NN(neural network)' is used without definition. The definition of 'NN' is firstly defined at line 138.

5. To verify the effectiveness and usefulness of the proposed algorithm, a comparative simulation or experiment is required.

Reviewer 2 Report

The paper deals with robust adaptive control based on deep neural networks that is used in an inverse system identification environment to approximate the inverse of a nonlinear plant. Although the contribution of this paper is interesting, I have the following comments:

1) How is the minimum phase system condition verified in a nonlinear system? because, it is a necessary condition to apply the proposed method.

2) In practical applications, the devices of a control systems have saturation, what happens if the controller has saturation? I consider it important to incorporate in the analysis, the case when there is saturation in the control law.

3) Although the robustness in the closed-loop control system is mentioned when the proposed method is applied, it is necessary to carry out a formal analysis to affirm this property.

Reviewer 3 Report

The authors have presented a robust adaptive inverse control of nonlinear dynamic systems using deep neural network. This work is well written, but there are many languages issues. The topic of the paper is of sure interest for this journal. Its improvement is suggested in terms of the following comments:

1. The number of Keywords seems a little more, please reduce to 4-5 phrases.

2. The main contributions should be described clearly in the introduction part, and  any limitation of this approach should also be included in the work.

3. Some simulation or theoretical comparisons must  be  performed  to  show  the effectiveness of  the proposed strategy with other strategies, one useful approach has been developed in the work 'active disturbance rejection control for delayed electromagnetic docking of spacecraft in elliptical orbits', which can be refered to improve the quality of this work.

4. I suggest the authors plot the error curves in figs. 4,6,8,10 to further analyze the results.

5. A proof-reading should be performed before resubmission.

Round 2

Reviewer 2 Report

Please include the comments made in the reply within the content of the paper.
